# Blood Biomarkers Reflect Dementia Symptoms and Are Influenced by Cerebrovascular Lesions

**DOI:** 10.3390/ijms26052325

**Published:** 2025-03-05

**Authors:** Taizen Nakase, Yasuko Tatewaki, Yumi Takano, Shuko Nomura, Hae Woon Baek, Yasuyuki Taki

**Affiliations:** Department of Aging Research and Geriatric Medicine, Institute of Development, Aging and Cancer, Tohoku University, 4-1 Seiryo Machi, Sendai 980-8575, Miyagi, Japanhae.woon.baek.d3@tohoku.ac.jp (H.W.B.); yasuyuki.taki.c7@tohoku.ac.jp (Y.T.)

**Keywords:** dementia, plasma biomarker, Alzheimer’s disease, cognitive impairment, neuropsychological symptom

## Abstract

Dementia blood biomarkers are becoming increasingly important. Various factors, such as ischemic lesions and inflammation, can influence the pathomechanism of dementia. We aimed to evaluate the effects of past stroke lesions on blood biomarkers (BMs). Following approval from the institutional ethics committee, patients who were admitted to the memory clinic and were consented to written documents were enrolled (*n* = 111, average [standard deviation] age: 74.5 [9.1] years-old). Brain magnetic resonance imaging, cognitive function, and neuropsychological symptoms were analyzed. The amyloid-β 42 (Aβ42)/Aβ40 ratio, phosphorylated tau181 (p-tau181), glial fibrillary acidic protein (GFAP), neurofilament light chain (NfL), and Aβ42/p-tau181 ratio were assessed as plasma BMs. The patients were diagnosed with Alzheimer’s disease (*n* = 45), mild cognitive impairment (*n* = 56), depression (*n* = 8), and subjective cognitive impairment (*n* = 4). Bivariate analysis exhibited that all measured BM indicators were significantly associated with cognitive decline in patients without past stroke lesions. Whereas the patients with stroke lesions presented a significant association only between GFAP and cognitive decline (*p* = 0.0011). Multiple regression analysis showed that NfL significantly correlated with cognitive decline only in patients without stroke lesions (r = 0.4988, *p* = 0.0003) and with delusion only in those with stroke lesions (r = 0.5492, *p* = 0.0121). Past stroke lesions should be addressed in the assessment of the correlation between blood biomarkers and cognitive decline in dementia patients.

## 1. Introduction

Recently, the development of disease-modifying medicines has progressed even in the sector of dementia treatment domain, becoming increasingly important to detect pathogenic proteins, such as amyloid-β (Aβ) and phosphorylated tau protein (p-tau), for determining appropriate medications for patients with Alzheimer’s disease (AD). Examinations of amyloid positron emission tomography (PET) and cerebrospinal fluid (CSF) are used for the detection of Aβ and p-tau; however, each test presents some limitations, such as high costs and high invasiveness. Thus, introducing a more straightforward method for evaluating AD pathology is necessary. Additionally, blood biomarkers have recently been developed [1,2]. Plasma levels of Aβ42 and neurofilament light chain (NfL) are reportedly associated with the risk of AD and all-cause dementia, respectively [3]. Plasma levels of p-tau181 and p-tau217 are reportedly effective markers for predicting AD pathophysiology [4,5]. Moreover, it was reported that cases of subjective cognitive decline with Aβ positivity exhibited a faster decline of cognitive function compared to those without Aβ [6]. Among patients with AD who are Aβ positive, those with higher plasma p-tau181 levels reportedly exhibited faster worsening of cognitive impairment [7].

Meanwhile, the worsening of AD pathology can be caused by not only the accumulation of Aβ but also various causal factors, such as chronic insufficiency of cerebral blood flow, chronic inflammation, and dysfunction of the glymphatic system [8,9]. In a super-aged society, such as Japan, the prevalence rate of cerebrovascular disease and atrial fibrillation has increased. Notably, both conditions are risk factors for dementia. Many patients with AD pathology reportedly presented coexisting cerebrovascular lesions [10]. That is, cerebrovascular lesions coexisting with AD pathology may contribute to the worsening of cognitive impairment. Hence, the deteriorating source of dementia lies within the complex system, necessitating a multifaceted assessment to evaluate the pathogenesis of dementia.

Therefore, the effect of cerebrovascular lesions coexisting with AD should be evaluated using blood biomarkers. To date, blood biomarkers, such as high-sensitivity C-reactive protein, pentraxin, and the neutrophil-to-lymphocyte ratio, have been used to assess the risk and severity of cerebrovascular disease [11]. However, these biomarkers do not indicate cognitive impairment nor neuropsychological symptoms.

Moreover, patients with dementia may sometimes present with neuropsychological symptoms, such as depression, agitation, hallucinations, and delusions, along with dementia progression. Because neuropsychological symptoms in patients with dementia can be caused by dysfunction of neuronal networks [12], blood biomarkers might be associated with the appearance of these symptoms. Since cerebrovascular lesions often occur in the brain white matter, evaluating the involvement of stroke lesions is necessary when exploring the association between blood biomarkers and clinical symptoms.

Therefore, we aimed to clarify how cerebrovascular lesions may affect the evaluation of the association of blood biomarkers with cognitive impairment and neuropsychological symptoms in patients on the AD pathology continuum.

## 2. Results

### 2.1. Background of All Patients

The clinical characteristics of the patients are summarized in Table 1. About 40% of patients had AD type dementia (ADD), and 50% presented amnestic mild cognitive impairment (aMCI). Although the average age was not significantly different among the patient groups, those with subjective cognitive impairment (SCI) and depression were relatively younger than those in the other groups. Inevitably, the Mini-Mental State Examination (MMSE) scores in the ADD and aMCI groups were significantly worse than those in the SCI and depression groups (ADD vs. aMCI, *p* = 0.0002; ADD vs. SCI, *p* = 0.0001; ADD vs. depression, *p* < 0.0001; aMCI vs. SCI, *p* = 0.0136; and aMCI vs. depression, *p* = 0.0057). The Alzheimer’s disease assessment scale (ADAS) score in the ADD group was significantly worse than in other groups (ADD vs. aMCI, *p* < 0.0001; ADD vs. SCI, *p* = 0.0105; and ADD vs. depression, *p* = 0.0004). The percentages of comorbid risk factors, such as hyperlipidemia, diabetes mellitus, smoking, and history of stroke, did not differ between the groups. Although the amount of HbA1c was not significantly different between the groups, those in the ADD and aMCI groups were about 6% and relatively higher than those in the SCI and depression groups. Among patients with previous stroke lesions, lacunar infarction was the most frequent subtype, followed by atherosclerotic infarction. Past stroke occurred approximately 6 years before cognitive impairment was observed. No significant association was observed between the dementia subtype and percentage of patients who took any antithrombotic medication. The percentage of patients with hypertension was significantly lower in the SCI group than in the other groups.

### 2.2. Correlation Between Blood Biomarker Level and Clinical Diagnosis

The plasma concentrations of each blood biomarker are shown in Table 2. The average levels of Aβ40 and Aβ42 were not different among the four groups. The average levels of p-tau181, NfL, and Glial fibrillary acidic protein (GFAP) were significantly higher in the ADD and aMCI groups. Ratios of Aβ42/Aβ40 and Aβ42/p-tau181 were significantly lower in the ADD and aMCI groups. These data presented typical statistical results.

### 2.3. Association Between Plasma Aβ Level and Amyloid PET

According to the Aβ analysis from 37 patients who were examined using amyloid PET, 30 patients exhibited positive results. Representative amyloid PET images of amyloid-positive and negative patients are shown in Figure 1A. Receiver operating characteristic (ROC) analysis of these patients indicated that the Aβ42/Aβ40 ratio, Aβ42/p-tau181 ratio, and plasma GFAP level exhibited significant sensitivity and specificity for amyloid positivity (*p* = 0.0309, AUC: 0.8095, sensitivity: 83.3%, specificity: 85.7%; *p* = 0.0351, AUC: 0.7048, sensitivity: 56.7%, specificity: 85.7%; and *p* = 0.0436, AUC: 0.7857, sensitivity: 60.0%, specificity: 100%, respectively) compared to other biomarkers (p-tau181: *p* = 0.3213; and NfL: *p* = 0.1460) (Figure 1B).

### 2.4. Association of Plasma Biomarker Level and Severity of Cognitive Impairment

Plasma p-tau181 concentration levels demonstrated a significant correlation with MMSE and ADAS scores (r = −1.976; *p* = 0.0004 and r = 1.027; *p* = 0.0179, respectively). Plasma NfL levels also exhibited a significant correlation with the ADAS score (r = 0.9415; *p* = 0.0081). Similarly, plasma GFAP levels correlated significantly with MMSE and ADAS scores (r = −5.521; *p* = 0.0160 and r = 6.811; *p* < 0.0001, respectively). The Aβ42/Aβ40 ratio was significantly correlated with MMSE and ADAS scores (r = 0.00097; *p* = 0.0033 and r = −0.00068; *p* = 0.0037, respectively). The Aβ42/p-tau181 ratio also demonstrated significant correlations with MMSE and ADAS scores (r = 0.00932; *p* < 0.0001 and r = −0.00500; *p* = 0.0005, respectively). When patients with and without past stroke lesions were analyzed separately, distinctive differences emerged (Figure 2). In patients without past stroke lesions, all measured plasma biomarker values exhibited significant correlations with the severity of cognitive impairment, as evaluated by MMSE and ADAS (Figure 2A–E: left columns). However, among patients with past stroke lesions, only GFAP exhibited a significant correlation with the ADAS score (Figure 2C, right column: r = 11.400, *p* = 0.0011).

### 2.5. Correlation Between Plasma Biomarkers and Cognitive Decline by Multivariate Analysis

All blood biomarker data were analyzed using MMSE or ADAS scores by multivariate analysis. Most of blood biomarker data exhibited a significant correlation with cognitive scores. No significant correlation was exhibited between the plasma NfL levels and MMSE scores (Table 3). Then, all data were analyzed separately for patients with and without previous stroke lesions (Table 4). Among those without stroke lesions, significant correlations were exhibited between the plasma concentration levels of all biomarkers and MMSE scores. Regarding the association with the ADAS score, only NfL exhibited a significant correlation (*p* = 0.0003). Among the patients with stroke lesions, no significant correlation was observed between plasma biomarker levels and MMSE scores. Only plasma GFAP levels showed a significant association with ADAS score (*p* = 0.0011).

### 2.6. Correlation Between Plasma Biomarkers and Neuropsychological Symptoms Using Multivariate Analysis

Neuropsychological symptoms were sub-classified into four groups: depression, irritability, hallucinations, delusions, and apathy (Table 5). The percentages of each symptom among all patients were 25.2%, 16.2%, 6.3%, 10.8%, and 24.3%, respectively. Multivariate analysis showed that plasma NfL levels significantly correlated with hallucinations and delusions (*p* = 0.0275 and 0.0014, respectively). The Aβ42/Aβ40 ratio demonstrated a significant association with depression (*p* = 0.0409).

All data were assessed separately for patients with and without previous stroke lesions (Table 6). Among those without stroke lesions, a significant correlation was observed between the Aβ42/Aβ40 ratio and depression (*p* = 0.0407). Among patients with stroke lesions, plasma NfL levels significantly correlated with delusions (*p* = 0.0121).

## 3. Discussion

Our results clearly indicated that not only can blood biomarkers be an effective indicator of cognitive decline in patients with dementia, but the existence of vascular lesions that may affect AD pathology could influence the results of measuring plasma concentration levels of NfL and GFAP.

Recently, various studies have demonstrated the effectiveness of plasma biomarkers in detecting Alzheimer’s pathology [2,3,4,5,7,13]. Moreover, prospective observational studies have reported that a decrease in the plasma Aβ42/Aβ40 ratio may predict AD pathology and increased levels of NfL and GFAP may serve as potential markers for all cause dementia [14,15]. Additionally, synaptic dysfunction, oligodendrocytic dysfunction, and glial activation have been observed prior to neuronal damage in patients with dementia [8,16,17]. It could be interesting to evaluate not only Aβ and p-tau pathology but also NfL and GFAP for revealing AD pathology with a complex background.

In this study, patients who were admitted to our memory clinic were consecutively requested to participate, and those with AD, aMCI, SCI, and depression were included. Among them, the AD and aMCI groups exhibited significantly higher serum concentrations of p-tau181, NfL, and GFAP. Ratios of Aβ42/Aβ40 and Aβ42/p-tau181 were significantly lower in these groups. Regarding this, plasma concentration of blood biomarkers was analyzed with the severity of cognitive impairment. We demonstrated significant correlations in almost all examined indicators except for plasma NfL concentration and MMSE score. This finding aligns with previous research data [4]. It suggests that blood biomarker measurement can support the assessment of cognitive decline severity. However, the signal–noise ratio of blood biomarkers may be lower than that of CSF biomarkers [18]. Potential contributing factors might include effects of peripheral nervous system disorders, liver and kidney dysfunction, or the influence of coexisting intracranial lesions in the central nervous system. Indeed, pathological studies have shown that cerebral white matter lesions can exacerbate AD pathology [19]. Furthermore, cognitive function in patients with an accompanying white matter lesion or ischemic lesions alongside AD is reportedly more severely impaired than that in patients with Alzheimer’s pathology alone [20,21]. Aβ42 and NfL as CSF biomarkers are reportedly associated with white matter hyperintensity lesions [22]. Consequently, patients with and without past stroke lesions were analyzed separately in this study. The significance of the correlation between blood biomarkers and cognitive decline diminished when patients with stroke lesions were analyzed separately. Only plasma GFAP level demonstrated a significant correlation with the ADAS score. Our results suggest that the presence of the past stroke lesions can affect the amount of GFAP, which may originate from gliosis or ischemic neuronal damage. In this study, since the number of patients on antithrombotic medications was very small, we could not assess the effect of such medication. Future studies must include evaluations of the association of medication with blood biomarker concentration levels.

Diabetes mellitus can play an important role in the progression of atherosclerosis, leading to the pathogenesis of white matter lesions. Moreover, altered insulin resistance was reportedly observed in the brain of diabetes patients, elderly persons, and AD patients [23]. Impaired insulin metabolism may cause neuronal damage; thus, it will be a risk factor of dementia. In our findings, patients with ADD and aMCI can be considered as being prediabetic status (HbA1c was between 5.7% and 6.4% [24]). It might provide an effective information in predicting cognitive decline to assess not only blood biomarkers but also severity of brain white matter lesion and insulin resistance, i.e., HbA1c level.

According to multiple regression analysis of blood biomarkers and neuropsychological symptoms, the plasma NfL concentration exhibited a significant correlation with hallucinations and delusions. Since hallucinations and delusions are reportedly influenced by white matter lesions [25], alterations in plasma NfL concentration may indicate damage to cerebral white matter. When patients with and without past stroke lesions were separately analyzed, this observation became more prominent in patients with stroke lesions. Aging and ischemic damage in the brain may impair white matter metabolism, contributing to the pathophysiology of neurodegenerative diseases, such as AD and Parkinson’s disease [26,27]. Another study reported that partial damage to the blood–brain barrier caused by ischemic stroke could trigger the effusion of white blood cells, platelets, cytokines, Aβ, etc. from blood vessels. These phenomena could lead to dysfunction of the peripheral circulation, initiation of inflammatory response, and the accumulation of Aβ [28]. Therefore, the evaluation of plasma NfL and GFAP levels must be included for assessing white matter pathology in the early stages of AD.

Our study has some limitations. First, the number of participants was relatively small, particularly the number of patients who underwent amyloid PET, which was only 37. Moreover, only a few numbers of patients presented neuropsychological symptoms, weakening the statistical power of the analysis. Based on this, it is necessary to consider the risk of overestimation. Further research is needed to validate our data using a larger cohort. Second, patients with depression were included in this study. Increasing the number of cases without dementia and separately evaluating them as a control group, which includes patients with depression, will be required. Third, the information about the distribution of past stroke lesions was not obtained, preventing us from investigating their lesional impact. Future studies should investigate not only the effect of stroke lesions but also their distribution. Additionally, we compared the blood biomarkers between patients with and without past stroke lesions. Regarding ischemic lesions, future studies should analyze blood biomarkers considering the influence of white matter lesions in addition to stroke lesions.

## 4. Materials and Methods

### 4.1. Patients

All procedures were approved by the Ethical Committee of Tohoku University Hospital (# 2022-1-385). All patients who were admitted to our memory clinic between August 2022 and October 2024 were asked to participate in this study and 128 patients provided written informed consent. Consequently, 111 patients met the inclusion criteria and were enrolled in this study (Figure 3). The inclusion criteria were admission to the memory clinic, with written consent, and undergoing routine clinical tests. The exclusion criteria included other diagnoses, such as vascular dementia, mixed type dementia, frontotemporal dementia, Lewy body dementia, Parkinson’s disease, corticobasal degeneration, epilepsy, severe kidney disease, hemodialysis, thyroid disease, major psychiatric diseases, active infectious disease, and active cancer.

Frontotemporal dementia, Lewy body dementia, Parkinson’s disease, corticobasal degeneration, and epilepsy were included as other diagnoses.

All patients underwent physical examination, blood sampling (for both laboratory tests and blood biomarker measurement), brain magnetic resonance imaging, and N-isopropyl-p-[^123^I] iodoamphetamine single photon emission computed tomography (^123^I-IMP SPECT) and were diagnosed by registered doctors. Patients with ADD, aMCI, SCI, and depression were included in this study. Clinical diagnosis and background characteristics were obtained using electronic patient records. Cognitive impairment was assessed using the MMSE and ADAS tests. Neuropsychological symptoms were evaluated using the Neuropsychiatric Inventory Questionnaire.

### 4.2. Biomarker Measurement

The blood sample was centrifuged at 4000 rpm at −20 °C, and the supernatant was corrected in a microtube and stored at −80 °C until analysis. The frozen samples were sent to the Advanced Neuroimaging Center, National Institute for Quantum Science and Technology (Chiba, Japan). In this facility, the plasma levels of Aβ40, Aβ42, p-tau181, NfL, and GFAP were analyzed with a single-molecule enzyme-linked immunosorbent assay technology using the p-Tau181 Advantage Kit v2.1 and the Neurology 4-Plex E assay kits (Simoa^®^, Quanterix, Billerica, MA, USA). A calibration curve was created according to the instrument manual, and the results of sample measurements were corrected [13,29].

Thirty-seven patients underwent amyloid PET using ^18^F flutemetamol (Nihon Medi-Physics Co. Ltd., Tokyo, Japan); alternatively, their CSF was examined.

### 4.3. Statistical Analysis

Data are presented as a number and percentage or as average ± standard deviation (SD). Comparisons of mean values and percentages between the groups were assessed using one-way analysis of variance and the chi-square test using Pearson’s method. The sensitivity and specificity of each blood biomarker against Aβ positivity in the amyloid PET or CSF study were evaluated by the ROC analysis. Linear regression analysis was used to assess the correlation between the plasma concentration of each blood biomarker and cognitive test scores, including MMSE and ADAS. Furthermore, multivariate analysis was used to investigate the relationship between blood biomarkers and MMSE and ADAS scores or the existence of neuropsychological symptoms. Brain vascular lesions reportedly affect AD pathology [9]; therefore, we analyzed data from patients with and without past stroke lesions. The JMP Pro (17.0.0, JMP Statistical Discovery LLC, Cary, NC) software was used for the analysis. Statistical significance was set at *p* < 0.05.

## 5. Conclusions

Blood biomarkers for the AD pathology continuum are possible indicators of worsening of cognitive impairment regardless of past stroke lesions. Notably, in patients with AD accompanied by past stroke lesions, plasma GFAP levels may be associated with the severity of cognitive impairment, and plasma NfL levels may reflect the severity of delusions as neuropsychological symptoms. In AD patients without past stroke lesions, plasma NfL levels may suggest the severity of cognitive impairment, and the plasma Aβ42/Aβ40 ratio may correlate with the severity of depression as neuropsychological symptoms. It can be said that not only Aβ and p-tau but other factors, such as GFAP and NfL, should be included in the assessment of blood biomarkers.

## Figures and Tables

**Figure 1 ijms-26-02325-f001:**
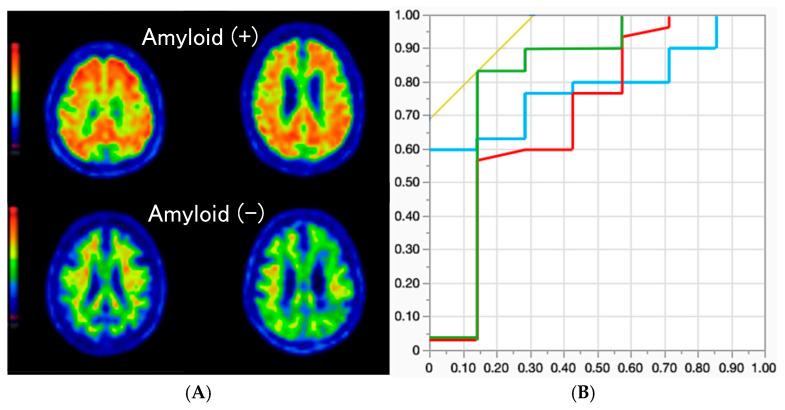
Representative amyloid PET images of patients who were amyloid-positive (top two images) and amyloid-negative (bottom two images) are depicted in (**A**). (**B**) presents ROC curves for the Aβ42/Aβ40 ratio (green), Aβ42/p-tau181 ratio (red), and plasma GFAP level (blue). The X and Y axes represent 1-specificity and sensitivity, respectively.

**Figure 2 ijms-26-02325-f002:**
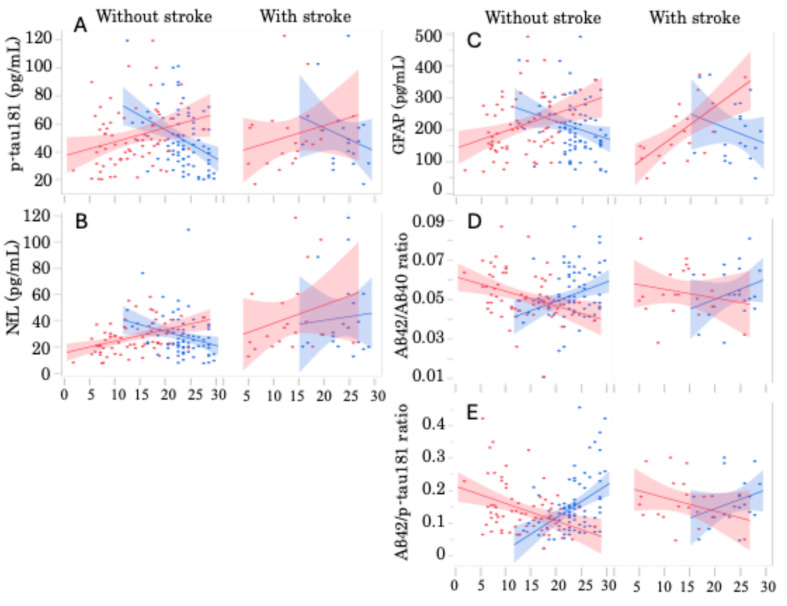
Correlation between plasma biomarker levels and cognitive decline in patients with and without stroke lesions. Scatter graphs of patients without stroke lesions (left) and with stroke lesions (right) exhibited the correlation of blood biomarkers of (**A**) p-tau181, (**B**) NfL, (**C**) GFAP, (**D**) Aβ42/Aβ40, and (**E**) Aβ42/p-tau181 with MMSE (blue) or ADAS (red) scores. Linear lines indicate the correlation lines. The formulae for each line are as follows: (**A**) y = 97.76 − 2.046x(MMSE), *p* = 0.0005, and y = 36.65 + 1.009x(ADAS), *p* = 0.0340 in patients without stroke, and y = 91.21 − 1.676x(MMSE), *p* = 0.2957 and y = 36.58 + 1.106x(ADAS), *p* = 0.3080 in patients with stroke; (**B**) y = 54.46 − 1.047x(MMSE), *p* = 0.0186 and y = 16.57 + 0.873x(ADAS), *p* = 0.0005 in patients without stroke, and y = 29.73 + 0.577x(MMSE), *p* = 0.7626 and y = 25.14 + 1.374x(ADAS), *p* = 0.2799 in patients with stroke; (**C**) y = 337.8 − 5.352x(MMSE), *p* = 0.0334 and y = 142.7 + 5.479x(ADAS), *p* = 0.0048 in patients without stroke, and y = 346.1 − 6.183x(MMSE), *p* = 0.2928 and y = 46.9 + 11.4x(ADAS), *p* = 0.0011 in patients with stroke; (**D**) y = 0.029 + 0.0096x(MMSE), *p* = 0.0089 and y = 0.061 − 0.001x(ADAS), *p* = 0.0047 in patients without stroke, and y = 0.0294 + 0.001x(MMSE), *p* = 0.2159 and y = 0.059 − 0.0004x(ADAS), *p* = 0.4227 in patients with stroke; and (**E**) y = −0.088 + 0.010x(MMSE), *p* < 0.0001 and y = 0.218 − 0.005x(ADAS), *p* = 0.0017 in patients without stroke, and y = 0.029 + 0.006x(MMSE), *p* = 0.2096 and y = 0.223 − 0.004x(ADAS), *p* = 0.1809 in patients with stroke.

**Figure 3 ijms-26-02325-f003:**
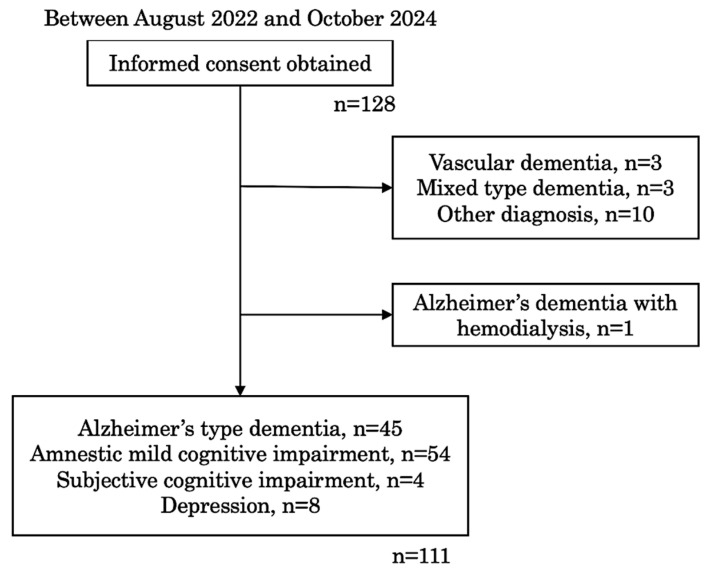
Flow chart of patients’ selection.

**Table 1 ijms-26-02325-t001:** Clinical characteristics.

	Total	ADD	aMCI	SCI	Depression	*p*
n	111	45	56	4	8	
Female (%)	62.2%	71.1%	53.7%	50.0%	75.0%	0.2629
Age (ave. ± SD years-old)	74.5 ± 9.1	74.8 ± 10.3	75.6 ± 7.0	68.5 ± 1.7	67.9 ± 13.1	0.0758
MMSE	23.8 ± 4.4	21.6 ± 4.6	24.6 ± 3.3	29.5 ± 0.6	28.6 ± 2.2	<0.0001
ADAS	13.6 ± 5.8	16.7 ± 5.7	12.0 ± 5.0	6.8 ± 1.1	7.6 ± 3.3	<0.0001
HT (%)	55.9%	53.3%	57.4%	0.0%	87.5%	0.0371
HL (%)	45.0%	42.2%	44.4%	50.0%	62.5%	0.7585
LDL (ave. ± SD mg/L)	109.6 ± 37.2	103.3 ± 35.9	117.1 ± 5.3	95.0 ± 23.8	101.1 ± 26.1	0.2600
DM (%)	20.7%	20.0%	22.2%	0.0%	25.0%	0.7475
HbA1c (ave. ± SD %)	6.0 ± 0.8	6.1 ± 1.0	6.0 ± 0.8	5.9 ± 0.3	5.7 ± 0.2	0.6056
Smoking (%)	20.7%	22.2%	20.4%	0.0%	25.0%	0.7529
BMI	22.4 ± 3.4	22.0 ± 3.7	22.3 ± 2.9	22.2 ± 2.0	24.6 ± 5.2	0.3056
Stroke (n, %)	20, 18.0%	7, 13.3%	12, 25.9%	1, 25.0%	0, 0.0%	0.1332
Subtype: embolic (n)	1	0	1	0	0	na
atherosclerotic (n)	4	2	2 (1)	0	0	na
lacunar (n)	11	4 (1)	7 (3)	0	0	na
Hemorrhage (n)	4	1	2	1	0	na
Duration (ave. ± SD years)	6.5 ± 3.2	7.3 ± 3.5	6.0 ± 3.1	5.0	0	0.4271

Numbers within parentheses in the stroke subtypes indicate the number of patients who are taking antithrombotic medication. ADD, Alzheimer’s disease type dementia; aMCI, amnestic mild cognitive impairment; SCI, subjective cognitive impairment; SD, standard deviation; MMSE, Mini-Mental State Examination; ADAS, Alzheimer’s Disease Assessment Scale; HT, hypertension; HL, hyperlipidemia; LDL, low-density lipoprotein; DM, diabetes mellitus; BMI, body mass index; and na, not applicable.

**Table 2 ijms-26-02325-t002:** Blood biomarker concentration in the patients’ groups.

ave. ± SD	ADD	aMCI	SCI	Depression	*p*
Aβ40 (pg/mL)	123.4 ± 23.9	122.6 ± 24.1	110.8 ± 14.3	102.2 ± 20.6	0.0942
Aβ42 (pg/mL)	6.12 ± 2.12	6.45 ± 1.72	7.91 ± 1.19	6.60 ± 2.14	0.3166
p-tau181 (pg/mL)	55.6 ± 19.2	45.6 ± 21.7	31.3 ± 16.6	32.5 ± 13.6	0.0032
NfL (pg/mL)	37.5 ± 25.0	30.7 ± 16.7	16.5 ± 44.9	17.7 ± 9.5	0.0216
GFAP (pg/mL)	227.6 ± 78.9	195.5 ± 89.2	125.0 ± 55.8	94.0 ± 32.8	0.0002
Aβ42/Aβ40	0.049 ± 0.013	0.053 ± 0.013	0.071 ± 0.005	0.064 ± 0.017	0.0007
Aβ42/p-tau181	0.121 ± 0.056	0.170 ± 0.088	0.299 ± 0.130	0.239 ± 0.111	<0.0001

ADD, Alzheimer’s disease type dementia; aMCI, amnestic mild cognitive impairment; SCI, subjective cognitive impairment; Aβ, amyloid-beta; p-tau181, phosphorylated tau protein 181; NfL, neurofilament light chain; and GFAP, Glial Fibrillary Acidic Protein.

**Table 3 ijms-26-02325-t003:** Correlation between plasma biomarker levels and MMSE or ADAS scores.

	MMSE		ADAS	
	r	*p*	r	*p*
p-tau181	−0.3485	0.0004	0.2561	0.0167
NfL	−0.1416	0.1600	0.2738	0.0103
GFAP	−0.2403	0.0160	0.3936	0.0002
Aβ42/Aβ40	0.2907	0.0033	−0.3015	0.0045
Aβ42/p-tau181	0.4159	<0.0001	−0.3335	0.0016

MMSE, Mini-Mental State Examination; ADAS, Alzheimer’s dementia assessment scale; p-tau181, phosphorylated tau protein 181; NfL, neurofilament light chain; GFAP, Glial Fibrillary Acidic Protein; Aβ42/Aβ40, ratio of amyloid-beta 42 to amyloid-beta 40; Aβ42/p-tau181, ratio of amyloid-beta 42 to phosphorylated tau protein 181.

**Table 4 ijms-26-02325-t004:** Correlation between plasma biomarker levels and MMSE or ADAS scores, separately analyzed in patients with and without past stroke lesions.

	Without Stroke				With Stroke			
	MMSE		ADAS		MMSE		ADAS	
	r	*p*	r	*p*	r	*p*	r	*p*
p-tau181	−0.3801	0.0005	0.2288	0.1139	−0.2460	0.2957	0.2401	0.3080
NfL	−0.2625	0.0186	0.4988	0.0003	0.0721	0.7626	0.2540	0.2799
GFAP	−0.2382	0.0334	0.2646	0.0662	−0.2475	0.2928	0.6749	0.0011
Aβ42/Aβ40	0.2906	0.0089	−0.2609	0.0702	0.2894	0.2159	−0.1898	0.4227
Aβ42/p-tau181	0.4399	<0.0001	−0.2289	0.1136	0.2932	0.2096	−0.3117	0.1809

MMSE, Mini-Mental State Examination; ADAS, Alzheimer’s dementia assessment scale; p-tau181, phosphorylated tau protein 181; NfL, neurofilament light chain; GFAP, Glial Fibrillary Acidic Protein; Aβ42/Aβ40, ratio of amyloid-beta 42 to amyloid-beta 40; Aβ42/p-tau181, ratio of amyloid-beta 42 to phosphorylated tau protein 181.

**Table 5 ijms-26-02325-t005:** Correlation between biomarkers and neuropsychological symptoms.

	Depression		Irritability		Hallucinations		Delusions		Apathy	
n	28		18		7		12		27	
	r	*p*	r	*p*	r	*p*	r	*p*	r	*p*
p-tau181	−0.0045	0.9644	0.0547	0.5891	0.0902	0.3722	0.0323	0.7493	0.0726	0.4726
NfL	−0.0084	0.9343	0.1513	0.1328	0.2204	0.0275	0.3159	0.0014	0.0627	0.5352
GFAP	−0.0182	0.8575	−0.0617	0.5421	0.0060	0.9526	0.0057	0.9551	0.0381	0.7066
Aβ42/Aβ40	0.2049	0.0409	−0.0647	0.5225	0.1443	0.1521	0.1386	0.1689	−0.0813	0.4215
Aβ42/p-tau181	0.0096	0.9242	−0.1118	0.2679	0.0149	0.8830	−0.0085	0.9333	0.0053	0.9581

p-tau181, phosphorylated tau protein 181; NfL, neurofilament light chain; GFAP, Glial Fibrillary Acidic Protein; Aβ42/Aβ40, ratio of amyloid-beta 42 to amyloid-beta 40; Aβ42/p-tau181, ratio of amyloid-beta 42 to phosphorylated tau protein 181.

**Table 6 ijms-26-02325-t006:** Correlation between biomarkers and neuropsychological symptoms in patients with and without stroke lesions.

Without Stroke Lesions										
	Depression		Irritability		Hallucinations		Delusions		Apathy	
n	23		13		4		9		22	
	r	*p*	r	*p*	r	*p*	r	*p*	r	*p*
p-tau181	0.0393	0.7290	0.0668	0.5561	0.1349	0.2328	0.0661	0.5601	0.1692	0.1336
NfL	−0.0103	0.9278	0.1384	0.2209	0.2079	0.0642	0.2067	0.0659	0.0530	0.6406
GFAP	0.0076	0.9470	−0.0160	0.8881	−0.0473	0.6767	−0.0160	0.8880	0.0395	0.7277
Aβ42/Aβ40	0.2294	0.0407	0.0082	0.9421	0.1633	0.1477	0.0931	0.4114	−0.0445	0.6950
Aβ42/p-tau181	−0.0181	0.8733	−0.0904	0.4252	−0.0084	0.9407	−0.0400	0.7248	−0.0680	0.5489
With stroke lesions										
n	5		5		3		3		5	
p-tau181	−0.1510	0.5251	0.0074	0.9753	−0.0131	0.9563	−0.0750	0.7534	−0.2399	0.3082
NfL	−0.0289	0.9039	0.1149	0.6296	0.1579	0.5061	0.5492	0.0121	0.0644	0.7875
GFAP	−0.1072	0.6528	−0.1832	0.4393	0.1629	0.4925	0.0912	0.7022	0.0429	0.8574
Aβ42/Aβ40	0.1051	0.6591	−0.3341	0.1499	0.0901	0.7056	0.2941	0.2081	−0.2313	0.3264
Aβ42/p-tau181	0.1318	0.5795	−0.2542	0.2794	0.0378	0.8744	0.1095	0.6459	0.3323	0.1523

p-tau181, phosphorylated tau protein 181; NfL, neurofilament light chain; GFAP, Glial Fibrillary Acidic Protein; Aβ42/Aβ40, ratio of amyloid-beta 42 to amyloid-beta 40; Aβ42/p-tau181, ratio of amyloid-beta 42 to phosphorylated tau protein 181.

## Data Availability

The datasets used and analyzed during the current study are available from the corresponding author upon reasonable request.

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
