# Peer review of "Blood Biomarkers Reflect Dementia Symptoms and Are Influenced by Cerebrovascular Lesions"

_ijms, 2025, doi:10.3390/ijms26052325_

Round 1

Reviewer 1 Report

Comments and Suggestions for Authors

This manuscript by Nakase T et al., evaluated the serum biomarkers to correlate the vascular lesion risk in AD patients. The authors plot of understanding and finding the correlation of ischemic lesions with any serum markers of Dementia, AD.

This manuscript posses multiple drawbacks;

1)  The title is misleading. Readers will be confused what authors trying to convey. Is it cerebral vascular lesions or coronary lesions. Is it the amyloid burden that cause cerebral lesions in the brain, increases with age and manifest as dementia, MCI and finally AD? what is the rationale for hypothesis. This study dosent stand its base without clearly stating their hypothesis.

2) Patients information is insufficient. Eg. Type of stroke, onset, duration, any treatment, smoking history, lipid profile- these are crucial for proper evaluations of the correlations, however, simply ignored in this study. 

3) Serum markers evaluations: This is another incomplete section. The authors should explain details of the type of test, kits used to measure the biomarkers.

Apart from the above mentioned criteria, this manuscript maybe not interesting for the broader readership as the quality of the manuscript is low as compared to the similar studies published already.

I apologies to say that my decision is not in favor.

Reviewer 2 Report

Comments and Suggestions for Authors

Title: Impact of vascular lesions on the blood biomarkers for Alzheimer’s disease.

Summary

In this manuscript, Taizen Nakase et al. evaluated the effects of past stroke lesions on blood biomarkers for Alzheimer’s disease. Amyloid beta 42 (Aβ42)/Aβ40 ratio, phosphorylated tau181 (p-tau181), glial fibrillary acidic protein (GFAP), neurofilament light chain (NfL), and Aβ42/p-tau181 ratio were measured in serum samples. The authors conclude that past stroke history should be considered when assessing blood biomarkers and that GFAP may be a useful biomarker for reflecting the severity of AD pathology in the presence of vascular lesions. However, the manuscript does not meet the standards for publication in the International Journal of Molecular Sciences. Below are major comments and suggestions for improving the manuscript:

1. The abbreviation BPSD should not be used in the keywords unless it is widely recognized in related literature. If it is not commonly used, it should also be removed from the main text.

2. The Introduction should be improved to provide more comprehensive background information and a clearer statement of the study’s aim.

3. Each subsection title should summarize the key conclusions rather than simply listing the names of experiments or assays.

4. Figures 1 and 2 contain error messages. These figures should be corrected and improved to ensure high-quality data presentation.

5. More details on biomarker measurements should be provided to enhance the study’s methodological robustness and validity.

6. The manuscript should be reorganized to align with the journal’s formatting and structural guidelines. Specifically, the Methods section should be placed before the Results section.

7. The Conclusion should be expanded to provide a stronger summary of the study’s findings. Alternatively, consider integrating the Conclusion with the Discussion for a more cohesive presentation.

8. Each figure contains limited data, weakening the overall impact of the findings. Additional experiments and more comprehensive data are needed to strengthen the validity of the study’s results.

Comments on the Quality of English Language

The English could be improved to more clearly express the research.

Reviewer 3 Report

Comments and Suggestions for Authors

Major points

- a major language revision by a native English speaker is mandatory for the entire manuscript

- the introduction is somewhat too short; in my opinion, a few further paragraphs could be added to the introduction

- results section 2.2 “According to the Aβ analysis from 37 patients who were examined by amyloid PET, 30 patients showed positive result.” – this whole paragraph should be added to a new section as it does not totally fit the section 2.2 entitled Blood biomarker analysis; Furthermore, consider adding some images from the PET scans and the ROC graph to this section

- results section 2.2 “Receiver operating characteristic (ROC) analysis based on these patients revealed that Aβ42/Aβ40 ratio showed most significant sensitivity and specificity for the amyloid positive”- it is worth pointing out that several other biomarkers were also statistically significant (e.g., GFAP: p=0.0436 and Aβ42/p-tau181: p=0.0351)

- Methods “and with other diseases which may affect blood test data” – this is far too vague; please specify the respective exclusion criteria

- Limitations – does the lack of a control group (e.g., no dementia) represent a limitation of the study?

- Table 3-5 – define the r value presented in the table. Presenting both r and p values in the same font somewhat decreases the readability of the table. In order to improve the readability of the table consider reformatting it

- the conclusions section should be redone as it does not fully encompass the results of the study

Minor points

- Figure 1 – a few words are underlined; The text presenting the contents and statistical results of the figure should be reformatted to clearly indicate that it is an integral part of the figure and not another paragraph of the manuscript

- The same also applies for Figure 2

- Figure 3 – there is enough space in order to fully write the abbreviated words

Comments on the Quality of English Language

- a major language revision by a native English speaker is mandatory for the entire manuscript

Round 2

Reviewer 1 Report

Comments and Suggestions for Authors

I appreciate the authors for addressing my concerns. This manuscript got benefitted from the peer review and can be accepted.

Author Response

I appreciate the authors for addressing my concerns. This manuscript got benefitted from the peer review and can be accepted.

Thank you very much for your effective advice. We were able to make our manuscript clearer owing to your appropriate comments.

Reviewer 2 Report

Comments and Suggestions for Authors

This manuscript is not publishable in the IJMS. 

1. The quality of data presentation should be improved.

2. Additional experiments and more comprehensive data analysis are needed to strengthen the validity of the study’s results.

Author Response

This manuscript is not publishable in the IJMS. 

1. The quality of data presentation should be improved.

2. Additional experiments and more comprehensive data analysis are needed to strengthen the validity of the study’s results.

Thank you for the reviewer’s comments. We carefully checked all data again and added results in the table 5 and 6. Then, we described limited point of our findings in the discussion section for making our results more accurate.

Reviewer 3 Report

Comments and Suggestions for Authors

The manuscript has been significantly improved during the first round of revisions. However, I would like to point out that some further changes are required.

- English language – the English writing has been significantly improved throughout the manuscript; however, some further revisions are necessary (e.g., “Among patients with stroke lesions, only GFAP was significantly associated (p=0.0011)”; “Although blood biomarkers are effective in assessing cognitive decline, past stroke lesions may affect the measurements.”; “exhibited a firster decline of cognitive function”; “Notably, both of these conditions are risk for dementia”; and others)

- Table 1 – most groups seem to have somewhat elevated HbA1c levels and could be considered as being prediabetic (doi: 10.4137/BMI.S38440.); the link between altered insulin resistance and neurological disorders would make a further interesting point for the discussion section

- In my opinion, the conclusions section could be slightly redone in order to fully encompass the difference between patients without stroke and with previous stroke; furthermore, another significant conclusion would be that that all examined plasma markers correlated with ADAS scores – this fact would prove helpful in the general examination of a dementia patient with unknown stroke history

Without stroke

  • Plasma NfL - severity of cognitive impairment
  • Plasma Aβ42/Aβ40 - severity of neuropsychological symptoms (depression)

With stroke

  • Plasma GFAP – severity of cognitive impairment
  • Plasma NfL – severity of neuropsychological symptoms (delusion)
Comments on the Quality of English Language

Further English improvements are required

Author Response

The manuscript has been significantly improved during the first round of revisions. However, I would like to point out that some further changes are required.

- English language – the English writing has been significantly improved throughout the manuscript; however, some further revisions are necessary (e.g., “Among patients with stroke lesions, only GFAP was significantly associated (p=0.0011)”; “Although blood biomarkers are effective in assessing cognitive decline, past stroke lesions may affect the measurements.”; “exhibited a firster decline of cognitive function”; “Notably, both of these conditions are risk for dementia”; and others)

Thank you very much for your precise comments and we are sorry for our poor English. We corrected some sentences following your comments. We also checked and corrected grammatical error throughout the text. (p1 line22-24, 26-28, p2 line45, 52-53, 64, 67, 79-80, p9 line292, 306, p10 line311, 313, and p11 line367)

- Table 1 – most groups seem to have somewhat elevated HbA1c levels and could be considered as being prediabetic (doi: 10.4137/BMI.S38440.); the link between altered insulin resistance and neurological disorders would make a further interesting point for the discussion section

Thank you for your effective advice. We agreed your precious comment and mentioned the difference of HbA1c levels among 4 groups in the results section (p2 line87- p3 line89). In the discussion section, we discussed the importance of impaired insulin metabolism in the dementia pathogenesis and mentioned prediabetic status of patients in the ADD and aMCI groups referring to a previous study as you introduced (p9 line284-291). 

- In my opinion, the conclusions section could be slightly redone in order to fully encompass the difference between patients without stroke and with previous stroke; furthermore, another significant conclusion would be that that all examined plasma markers correlated with ADAS scores – this fact would prove helpful in the general examination of a dementia patient with unknown stroke history

Without stroke

  • Plasma NfL - severity of cognitive impairment
  • Plasma Aβ42/Aβ40 - severity of neuropsychological symptoms (depression)

With stroke

  • Plasma GFAP – severity of cognitive impairment
  • Plasma NfL – severity of neuropsychological symptoms (delusion)

Thank you for your great advice. We reorganized the conclusion section. Following your comments, first, we wrote the effective point of our findings in which blood biomarkers can be possible indicators of worsening of cognitive impairment regardless of past stroke lesions. Then, we mentioned the difference of specific blood biomarkers which will correlate to the severity of symptoms between patients with and without past stroke lesions. (p11 line372-380)